# An Anti-CD7 Antibody–Drug Conjugate Target Showing Potent Antitumor Activity for T-Lymphoblastic Leukemia (T-ALL)

**DOI:** 10.3390/biom14010106

**Published:** 2024-01-15

**Authors:** Shiqi Wang, Ruyuan Zhang, Kunhong Zhong, Wenhao Guo, Aiping Tong

**Affiliations:** 1State Key Laboratory of Biotherapy and Cancer Center, West China Hospital, West China Medical School, Sichuan University, Chengdu 610041, China; wangsq@niicpm.com (S.W.); ru_yuan_zhang@163.com (R.Z.); 2Department of Neurosurgery, West China Hospital, Sichuan University, Chengdu 610041, China; zhongkunhong1993@126.com

**Keywords:** CD7, mAbs, ADC, T-ALL, therapy

## Abstract

Acute T-lymphoblastic leukemia (T-ALL) is a type of leukemia that can occur in both pediatric and adult populations. Compared to acute B-cell lymphoblastic leukemia (B-ALL), patients with T-cell T-ALL have a poorer therapeutic efficacy. In this study, a novel anti-CD7 antibody–drug conjugate (ADC, J87-Dxd) was successfully generated and used for T-ALL treatment. Firstly, to obtain anti-CD7 mAbs, we expressed and purified the CD7 protein extracellular domain. Utilizing hybridoma technology, we obtained three anti-CD7 mAbs (J87, G73 and A15) with a high affinity for CD7. Both the results of immunofluorescence and Biacore assay indicated that J87 (KD = 1.54 × 10^−10^ M) had the highest affinity among the three anti-CD7 mAbs. In addition, an internalization assay showed the internalization level of J87 to be higher than that of the other two mAbs. Next, we successfully generated the anti-CD7 ADC (J87-Dxd) by conjugating DXd to J87 via a cleavable maleimide-GGFG peptide linker. J87-Dxd also possessed the ability to recognize and bind CD7. Using J87-Dxd to treat T-ALL cells (Jurkat and CCRF-CEM), we observed that J87-Dxd bound to CD7 was internalized into T-ALL cells. Moreover, J87-Dxd treatment significantly induced the apoptosis of Jurkat and CCRF-CEM cells. The IC50 (half-maximal inhibitory concentration) value of J87-Dxd against CCRF-CEM obtained by CCK-8 assay was 6.3 nM. Finally, to assess the antitumor efficacy of a J87-Dxd in vivo, we established T-ALL mouse models and treated mice with J87-Dxd or J87. The results showed that on day 24 after tumor inoculation, all mice treated with J87 or PBS died, whereas the survival rate of mice treated with J87-Dxd was 80%. H&E staining showed no significant organic changes in the heart, liver, spleen, lungs and kidneys of all mice. In summary, we demonstrated that the novel anti-CD7 ADC (J87-Dxd) had a potent and selective effect against CD7-expressing T-All cells both in vitro and in vivo, and could thus be expected to be further developed as a new drug for the treatment of T-ALL or other CD7-expression tumors.

## 1. Introduction

ALL is a hematological malignancy characterized by uncontrolled proliferation of abnormal, immature lymphocytes and their progenitors which ultimately leads to the replacement of bone marrow elements and other lymphoid organs. ALL is seen in patients with blastic transformation of B and T cells; it is the most common leukemia in the pediatric population, accounting for up to 80% of cases in this group vs. 20% of cases in adults. However, adults with ALL fare worse than their pediatric counterparts, with an overall 5-year survival rate of 40% in newly diagnosed patients, and only 10% for those with relapsed disease [1]. ALL comprises B- or T-lymphoblastic leukemia (B-ALL or T-ALL) and can be divided into three principal categories: B-lymphoblastic leukemia not otherwise specified, B-lymphoblastic leukemia with recurrent cytogenetic alterations and T-lymphoblastic leukemia [2]. B-ALL is the most common form of ALL, comprising >20 subtypes of variable prevalence according to age. By contrast, T-ALL is an uncommon form, accounting for only approximately one-fourth of ALL cases [3]. In recent decades, B-ALL treatments have improved dramatically, but success has been more limited in T-ALL. In particular, relapsed and refractory T-ALL cases remain extremely challenging to treat and those who cannot tolerate intensive treatment continue to have poor outcomes [4]. Although various therapeutic modalities, including chemotherapy, CAR-T therapy and bone marrow transplantation are applied to T-ALL, most of them have serious adverse effects or restrictions of use [5,6]. Therefore, it is still essential to develop new treatment options for T-ALL.

ADCs, in which a mAb is conjugated to biologically active drugs through chemical linkers, have emerged as a promising class of anticancer treatment agents, being one of the fastest growing fields in cancer therapy [7]. As of June 2023, thirteen ADCs have been approved by the Food and Drug Administration (FDA, Silver Spring, MD, USA) and are on the market. These drugs have been added to the therapeutic armamentarium of acute myeloblastic and lymphoblastic leukemias, various types of lymphoma, breast, gastric or gastroesophageal junction, as well as lung, urothelial, cervical, and ovarian cancers [8]. They have proven to deliver more potent and effective antitumor activities than standard practice in a wide variety of indications [9]. Besponsa^®^ (inotuzumab ozogamicin), which was approved by FDA in 2017, is generated by conjugating a humanized anti-CD22 monoclonal antibody to the cytotoxic antibiotic agent calicheamicin via a hydrazone [10]. CD22, a type I transmembrane glycoprotein, is ubiquitously expressed in the B-cell lineage. After administration of Besponsa^®^, the anti-CD22 monoclonal antibody binds to CD22 on B cells in B-ALL and is internalized into the cell, where the hydrazone linker undergoes hydrolysis to release calicheamicin, which causes DNA damage and cell death [11,12]. CD7, similar to CD22, is also a type I transmembrane glycoprotein and a member of the immunoglobulin superfamily. Normally, CD7 is expressed on human T and natural killer cells and on cells in the early stages of T-, B-, and myeloid cell differentiation, acting as a T-cell-associated antigen. Previous studies have demonstrated that CD7 is also expressed in over 95% of T-ALL, 30% of acute myeloid leukemia (AML), and some lymphomas [13,14,15,16]. Due to its widespread distribution on the tumors, CD7 is considered an attractive target antigen for immunotherapy of T-ALL.

As is well known, ADCs are complex targeted agents composed of three main components: a monoclonal antibody (mAb), a cytotoxic drug and a linker. By providing a selective targeting mechanism for cytotoxic drugs, ADCs improve the therapeutic index in clinical practice [17]. Here, we generated a novel anti-CD7 ADC (J87-Dxd) with an anti-CD7 mAb (J87), a topoisomerase I inhibitor (Deruxtecan, DXd) and a cleavable maleimide-GGFG peptide linker. Moreover, we demonstrated its potent antitumor activity against CD7-expressing T-ALL cells both in vitro and in vivo. With further development, this anti-CD7 ADC could be an effective drug for the treatment of T-ALL.

## 2. Materials and Methods

### 2.1. Mice

Seven-week-old immunodeficient NCG female mice were purchased from GemPharmatech LLC (Nanjing, China). Mice were maintained under specific pathogen-free conditions, and all procedures met the requirements of the National Institutes of Health and Institutional Animal Care and Use Committee. The animal experiments were approved by the West China Hospital of Sichuan University Biomedical Ethics Committee (ethical approval document: 20230426002).

### 2.2. Cell Lines and Culture Conditions

All tumor cell lines used in this study were originally purchased from the American Type Culture Collection (ATCC). CCRF-CEM-luciferase and Hela-CD7-GFP cell lines were produced by transducing lentiviral vectors that encoded the luciferase reporter gene or fusion gene CD7-GFP into the original cell lines. All cells were cultured in Dulbecco’s modified Eagle’s medium or RPMI-1640 (Gibco, Billings, MT, USA), supplemented with 10% fetal bovine serum, 2 mM l-glutamine (Invitrogen, Waltham, MA, USA) and 1 mM penicillin streptomycin combination (HyClone, Logan, UT, USA).

### 2.3. Expression and Purification of Recombinant Proteins

The gene sequence encoding human truncated CD7 (extracellular domain of CD7) was synthesized by GENERAL BIOL (Chuzhou, China), and was sub-cloned into a pVAX1-based expression vector with 6× His tag fusion or mFc at the C-terminus. Transient expression in the HEK293T cell line was performed by using expression vectors with the help of PEI. The cells were subsequently cultured in FreeStyle™ 293 Expression Medium (Thermo Fisher Scientific, Waltham, MA, USA) for 4~5 days. Then, the culture supernatants were harvested and centrifuged for 30 min at 10,000× *g*, 4 °C, followed by filtration through a 0.45 µm membrane. The recombinant proteins were initially purified by Ni-NTA column chromatography, and the recombinant proteins were analyzed through SDS-PAGE. Finally, the recombinant proteins were stored at −80 °C.

### 2.4. Generation of Monoclonal Antibody

Seven-week-old female BALB/c mice were immunized with the purified CD7 recombinant protein. Hybridomas were produced by fusion of spleen cells with SP2/0 cells. Hybridomas were screened using an ELISA, immunofluorescence, and flow cytometry. The mFc domain of the mouse anti-CD7 mAbs were replaced with the hFc domain derived from human IgG1 using genetic engineering. The recombinant mAbs were expressed and purified as described above. The purified mAbs were analyzed through reduced- or non-reduced SDS PAGE. The binding affinity of mAbs to CD7 recombinant protein was measured using Biacore ×100 instrument with a CM5 sensor chip (GE Healthcare, Chicago, IL, USA), according to a previously published procedure [18]. CD7-mFc was immobilized on CM5 using amine coupling (Amine coupling kit, GE Healthcare). The anti-CD7 mAbs were injected across the chip in a 2-fold dilution series. The equilibrium dissociation constant was obtained by using BIA evaluation 2.0 software. Next, the anti-CD7 mAbs were used as primary antibody to incubate CCRF-CEM cells for 0 h, 1 h, 2 h, 4 h and 6 h, respectively. The CCRF-CEM cells were then stained with FITC-conjugated second antibody, and the flow cytometric assay was performed to assess the levels of internalization.

### 2.5. Preparation of CD7-Dxd ADCs

Deruxtecan, a compound consisting of a DX-8951 (Dxd) derivative and a linker maleimide-GGFG peptide, was obtained from MedChemExpress (New Jersey, NJ, USA). The conjunction of an anti-CD7 mAb antibody to Deruxtecan was achieved by sulfhydryl coupling according to the manual. The anti-CD7 mAb (J87) concentration was quantified using the NanoDrop 2000 spectrophotometer (Thermo Fisher Scientific), and the concentration was adjusted to 5 mg/mL. Then, 10 mg of antibody was added into 114.66 μL of TCEP-HCl solution and kept at 37 °C for 3 h. Subsequently, 41.36 μL of Deruxtecan solution and 193.6 μL of DMSO were added and centrifuged at 40 rpm for 3 h at room temperature. The upper layer was desalted with ADC Preservation Buffer, filtered through a 0.22 μm cartridge, and the protein concentration was determined using a NanoDrop 2000 and stored at −80 °C.

### 2.6. Cell Apoptosis Assay

Cells were plated at 2 × 10^5^ cells/well and treated with 20 nM of J87-Dxd for 48 h at 37 °C. Following the treatment, cells were adjusted to 1 × 10^5^ cells per staining reaction. Cells was incubated in 100 μL of buffer containing 5 μL of FITC Annexin-V (Meiluncell^®^, Dalian, China) and 5 μL of 7-AAD (Meiluncell^®^) for 15 min. Finally, cells were resuspended in 400 μL of binding buffer and processed to flow cytometric analysis without washing using a BD Fortessa flow cytometer and analyzed using FlowJo 10.6.0 software.

### 2.7. Viability Assays

Cell viability was measured using the CCK-8 assay (BioLegend, San Diego, CA, USA). Cells were cultured at 5 × 10^3^ cells/well in 100 μL of growth medium in 96-well plates and incubated with J87-Dxd for 72 h. For viability assay, 10 μL of CCK-8 reagent was added to each well and mixed by gentle shaking. The plate was incubated at 37 °C for 2 h, and the absorbance was detected with a microplate reader at 450 nm. A previously generated anti-SLC3A2 ADC was used as the control ADC. The control ADC could bind to CCRF-CEM but not to Raji.

### 2.8. In Vivo Antitumor Analysis

A total of 15 NCG immunodeficient female mice aged 7 weeks and weighing approximately 20 g were used for the study. A total of 3 × 10^6^ CCRF-CEM-luciferase cells were implanted in mice with caudal vein injection. Nine days later, the mice were divided into three groups and treated with PBS, J87 or J87-Dxd (6 mg/kg), respectively. To monitor tumor growth, the In Vivo Imaging System (IVIS) (Caliper Life Sciences, Hopkinton, MA, USA) was used to record mice bioluminescence imaging. Bioluminescence was activated 10 min after intraperitoneal injection with 150 mg/kg d-luciferin (Beyotime, Shanghai, China). Living Image software^®^ 4.5.5 (PerkinElmer, Waltham, MA, USA) was used to analyze the data. During this period, the body weights of the mice were measured and recorded every 3 days. After the death of mice, the major organs (heart, liver, spleen, lung and kidney) were embedded for H&E staining.

### 2.9. Statistical Analysis

The data were visualized on graphs and presented as mean ± standard deviation (SD). The data were analyzed using GraphPad Prism software v.8.0. For Kaplan–Meier overall survival analysis, a log-rank test was used to compare each of the arms. Statistically significant differences were evaluated by Student’s *t*-test comparing two experimental groups. * *p* < 0.05 was considered to indicate a significant difference.

## 3. Results

### 3.1. Generation and Characterization of mAbs against CD7

The new anti-CD7 mAbs with high affinity were prepared by the traditional hybridoma technique. First, the extracellular domain of CD7 and CD7-mFc were successfully expressed and purified from HEK293T. The SDS-PAGE analysis of the purified protein showed that the bands located at approximately 27 kDa (CD7) and 60 KDa (CD7-mFc), which is consistent with the expected (Figure 1A). Next, the CD7 protein was used to immunize BALB/c mice to prepare the anti-CD7 mAbs. Using hybridoma technology and genetic engineering, we successfully isolated three highly potent clones that produced anti-CD7 mAbs with hFc (A15, G73 and J87). As shown in Figure 1B, reduced samples of the anti-CD7 mAbs yield heavy chains of approximately 50 kDa and light chains of approximately 25 kDa on SDS-PAGE. When analyzed with non-reduced SDS PAGE, the mAbs yield a single bond with a size of approximately 150 kDa. Subsequent immunofluorescence assay with the three mAbs as the primary antibody indicated that the mAbs were able to efficiently recognize and bind the CD7 protein on CD7-ovrexpressing Hela cells (Figure 1C). Finally, the values of the affinity constants (kd, ka, and KD) were determined by Biocore assay. The results showed that the three anti-CD7 mAbs exhibited high affinity for CD7 (KD = 10^−10^ M), with J87 showing the highest affinity (KD = 1.54 × 10^−10^ M) (Figure 1D and Table 1).

### 3.2. Internalization Assay of Anti-CD7 mAbs

After antigen binding, the internalization of the ADC-antigen complex is thought to be a crucial step in payload delivery for many ADCs. Thus, we assessed the level of the three anti-CD7 mAbs internalization in this part. Flow cytometric analysis with a PE anti-human CD7 antibody (BioLegend) showed that CD7 was highly expressed on the cell surface of T-ALL cell lines CCRF-CEM and Jurkat but not on the Burkitt lymphoma cell line Raji (Figure 2A). As expected, CCRF-CEM cells were also successfully stained by J87, G73 and A15 (Figure 2B). In order to assess the internalization levels of the mAbs, J87, G73 and A15 were used as primary antibodies, respectively, to incubate CCRF-CEM cells for the indicated lengths of time (Figure 2C). The treated CCRF-CEM cells were then stained with a FITC anti-human IgG Fc secondary antibody (BioLegend), and flow cytometric was performed to assess the levels of internalization. The result indicated that J87, G73 and A15 were progressively internalized in CCRF-CEM cells with the prolongation of mAbs incubation time, and the internalization level of J87 was higher than that of the other two mAbs.

### 3.3. Generation and Cytotoxicity Assessment of J87-Dxd

To generate J87-Dxd, J87 was treated with TCEP to reduce interchain disulfide bonds to expose the sulfhydryl group, and then Deruxtecan was added to react with the sulfhydryl group, and finally coupled and bound to form J87-Dxd (Figure 3A). CCRF-CEM cells were incubated with J87-Dxd, followed by staining with a FITC anti-human IgG Fc secondary antibody. Like J87, J87-Dxd specifically bonds to CD7 molecules on the surface of CCRF cells (Figure 3B). Flow cytometric analysis showed that J87-Dxd-treated cells had reduced levels of cell-surface CD7 molecule expression (Figure 3C). After 48 h of treatment with J87-Dxd, approximately 60% of Jurkat cells and approximately 80% of CCRF-CEM cells underwent apoptosis (Figure 3D,E).

### 3.4. In Vivo Antitumor Efficacy of J87-Dxd in T-ALL Models

To build T-ALL models, we purchased 7-week-old female NCG-immunodeficient mice and inoculated them with CCRF-CEM cells stably expressing luciferase. Six days after tumor inoculation, 15 mice were randomly divided into three groups and each group contained 5 mice. On day 9, the three groups were treated with a single intraperitoneal injection of PBS, J87 and J87-Dxd respectively. Tumor burden was monitored by the in vivo imaging system every 3 days (Figure 4A). As shown in Figure 4B, mice in the PBS or J87 group showed rapid progression of T-ALL, with four of five mice (PBS group) dying and two of five mice (J87 group) dying on day 21 after tumor inoculation. All of the mice in both the PBS and J87 groups died on day 24. By contrast, T-ALL progression was significantly inhibited in mice treated with J87-Dxd, and only one mouse in the group died 24 days after tumor inoculation (Figure 4B). In addition, weight loss began on day 12 in the PBS and J87 groups, but on day 17 in the J87-Dxd group (Figure 4C). This led to a significant survival advantage in the J87-Dxd group compared with mice in the other two groups (Figure 4D). H&E staining (40×) of the major organs in the mice was performed for the evaluation of the safety of J87-Dxd. As shown in Figure 4E, there were no significant organic changes in the heart, liver, spleen, lungs and kidneys of mice in each group.

## 4. Discussion

T-ALL is a disease derived from the uncontrolled proliferation of mature or immature T cells, with high rates of disease relapse and a poor prognosis [19]. Current treatments with intensive chemotherapy protocols and allogeneic bone marrow transplantation have shown a cure rate of 75% in pediatric patients and 50% in adults with T-ALL [20,21]. However, after induction and consolidation chemotherapy, approximately 30% of adult patients have minimal residual disease, which is likely the most important risk factor for relapse in T-ALL. Furthermore, T-ALL is incurable for the majority of relapsed patients, with the overall survival rate being less than 10% [22]. Allogeneic bone marrow transplantation presents significant limitations, such as limited availability, higher cost, and graft versus host disease. In addition, the outcome of the consolidation treatment is patient-dependent [23,24]. Therefore, developing alternative methods to address these challenges for treatment seems crucial.

Anticancer immunotherapies have emerged as new therapeutic pillars within cancer treatment, among which ADCs are a promising class of immunotherapies with the potential to specifically target tumor cells and ameliorate the therapeutic index of cytotoxic drugs. CD7 is a type I transmembrane glycoprotein with a MW of 40-kD, whose expression level has been found to be significantly upregulated in T-ALL cells compared to normal CD7-positive T cells. Thus, CD7 is considered as an attractive target for T-ALL immunotherapy [13,25]. In recent decades, studies have focused on using anti-CD7 CAR T cells or anti-CD7 antibodies to treat T-ALL [26,27,28,29,30]. However, since CD7 is also extensively expressed in normal T cells and natural killer cells, extending the success of CAR-T therapy to T cell malignancies faces challenges such as CAR-T cell fratricide, high production cost, long lagging time and potential product contaminations [30]. In addition, anti-CD7 mAbs alone showed weak inhibition of tumor growth [25,26].

Dxd, a potent DNA topoisomerase I inhibitor, has been used to generate an anti-HEAR2 ADC (Trastuzumab deruxtecan, T-Dxd). On 5 August 2022, T-Dxd was approved by the U.S. Food and Drug Administration for the treatment of HEAR2-positive metastatic breast cancer. In this study, a novel anti-CD7 ADC (J87-Dxd) was successfully generated by conjugating Dxd to J87 via a cleavable maleimide-GGFG peptide linker. Our data demonstrated that J87-Dxd had a potent and selective effect against T-ALL cells both in vitro and in vivo. Utilizing hybridoma technology, we obtained three novel anti-CD7 mAbs (J87, G73 and A15) with a high affinity for CD7. Among the three mAbs, J87 was selected and used for conjugating to Dxd because of its high affinity (KD = 1.54 × 10^−10^ M) for CD7 and high level of internalization. Moreover, CCRF-CEM cells could be successfully stained with J87-Dxd, indicating that J87-Dxd kept the ability of recognizing and binding CD7 on the cell surface. Flow analysis showed that J87-Dxd treatment decreased the expression level of CD7 proteins on the surface of Jurkat and CCRF-CEM cells. The phenomenon demonstrated that J87-Dxd could be internalized into Jurkat and CCRF-CEM cells. Meanwhile, J87-Dxd also triggered apoptosis in Jurkat (~60%) and CCRF-CEM cells (~80%) at a concentration of 20 nM. After 48 h of treatment, the IC50 values of J87-Dxd in CCRF-CEM and Raji cells were 6.372 nM and 338.5 nM, respectively (Appendix A). This result can be explained by the fact that J87-Dxd was able to bind to CCRF-CEM cells but not to Raji cells. Next, J87 or J87-Dxd was administered to CCRF-CEM mouse models by intraperitoneal injection. The results showed that on day 24 after tumor inoculation, all mice treated with J87 died, whereas the survival rate of mice treated with J87-Dxd was 80%. Thus, in the in vivo study, J87-Dxd treatment significantly inhibited the growth of T-ALL compared with J87 or PBS treatment. Subsequently, H&E staining also showed no significant organic changes in the heart, liver, spleen, lungs and kidneys of the J87- or J87-Dxd-treated mice. Although our data showed that J87-Dxd could efficiently and specifically kill CD7-expressing tumor cells in vitro and in vivo, there were still limitations of our study. First, only three cell lines (Jurkat, CCRF-CEM and Raji) were used to assess the specific antitumor activity of J87-Dxd. J87-Dxd should also be assayed for killing activity against other cells, such as tumor cells (MOLT-4 and RPMI-8226) and normal cells (T and NK). Second, although tumor growth was significantly inhibited compared with PBS or J87 injection, all mice injected with J87-Dxd still died 30 days after inoculation with CCRF-CEM cells. Thus, the administered dose of J87-Dxd still needs to be further optimized. To exclude the possibility that these antitumor effects were generated by free Dxd, we also performed an experiment to compare the survival time of CCRF-CEM mice treated with Dxd alone with PBS vehicle control. As a result, no significant difference was observed between the Dxd- and PBS-treated mice (Appendix A). Third, using the HIC method, we observed that J87-Dxd had a low DAR value (approximately 1), which likely reduced its antitumor effect (Appendix A). Fourth, NCG mice used here are triple immunodeficient and lack functional/mature T, B and NK cells. To closely mimic the human immune system, NCG mice should be humanized by engrafting human PBMCs or CD34 stem cells.

Here, we could produce the anti-CD7 mAbs by a eukaryotic expression system, and the mAbs were used to conjugate a cytotoxic agent (Dxd). While the J87-Dxd significantly inhibited and killed T-ALL cells in vitro or in vivo, the antitumor effects still need to be improved further. Using the anti-CD7 mAbs to conjugate other cytotoxic agents may show a more potent antitumor activity against CD7-expressing tumors. Overall, we demonstrated that the novel anti-CD7 ADC (J87-Dxd) could efficiently and specifically kill T-ALL cells in vitro and in vivo. Here, although we only emphasized the antitumor activity of J87-Dxd against T-ALL cells, it is likely that J87-Dxd remains effective against other CD7-positive tumor cells. Therefore, we identified that CD7 can serve as a target of ADC for T-ALL treatment and that J87-Dxd has potential to be further developed as a new drug for the treatment of T-ALL or other CD7-expressing tumors.

## Figures and Tables

**Figure 1 biomolecules-14-00106-f001:**
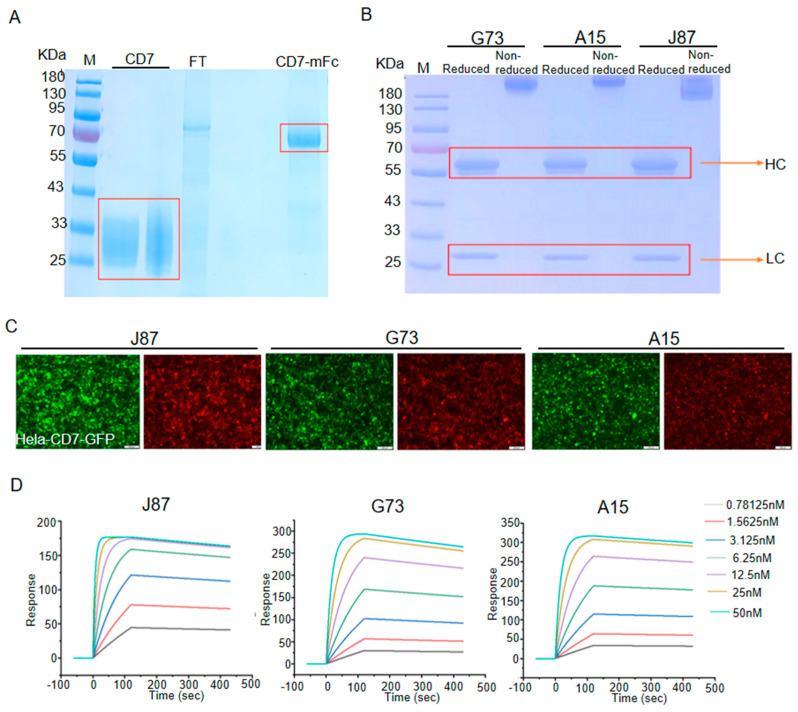
Generation and characterization of mAbs against CD7 (**A**) and (**B**) SDS−PAGE analysis for purified proteins. FT represents flow through, HC represents heavy chain, LC represents light chain. (**C**) Representative images of immunofluorescence staining using indicated mAbs as the primary antibodies. Scale bar = 100 μm. (**D**) The affinity of indicated mAbs binding to CD7 was determined on a Biacore T100 instrument.

**Figure 2 biomolecules-14-00106-f002:**
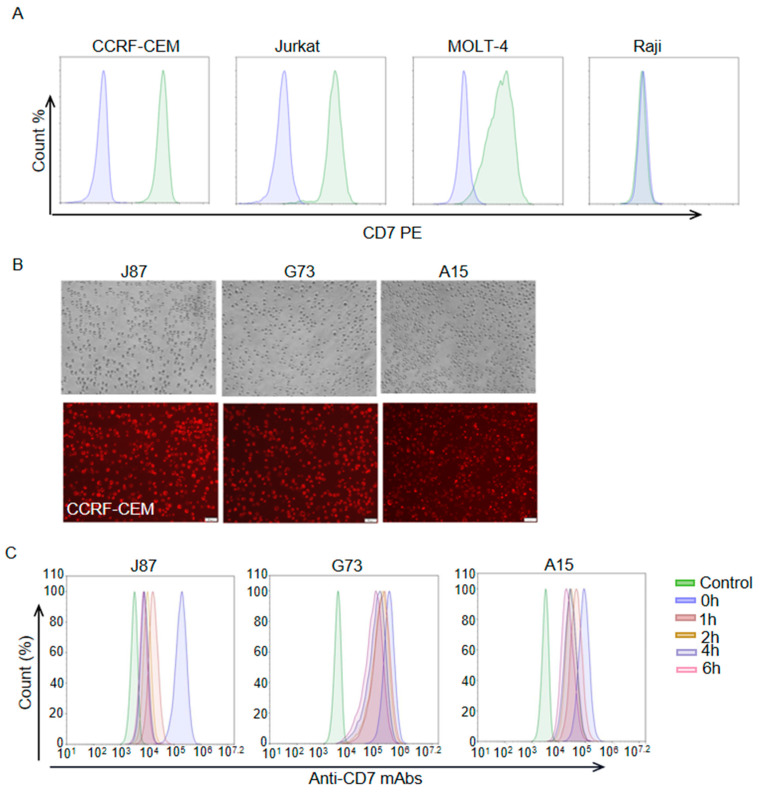
Internalization assessment of anti–CD7 mAbs. (**A**) Flow cytometry analysis of the expression levels of CD7 on cell surface of indicated cells. Blue color represents cells stained with PBS and green color represents cells stained with PE anti-human CD7 antibody. (**B**) Representative images of immunofluorescence staining using indicated mAbs as the primary antibodies and Alexa Fluor^®^ 594 anti-human IgG antibody as second antibodies. Scale bar = 50 μm. (**C**) Flow cytometry analysis of mAb internalization at indicated time point. The CCRF–CEM cells only stained with a FITC anti–human IgG Fc secondary antibody served as control.

**Figure 3 biomolecules-14-00106-f003:**
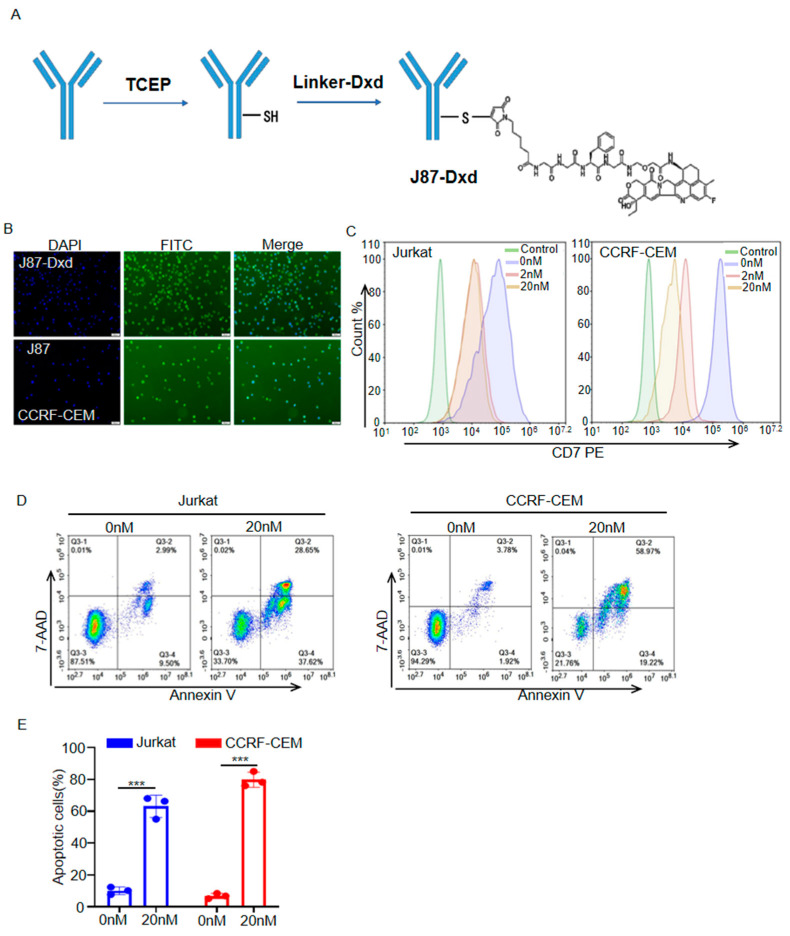
Generation and cytotoxicity assessment of J87–Dxd. (**A**) Schematic diagram of J87–Dxd preparation. (**B**) Representative images of immunofluorescence staining using J87–Dxd or J87 and a FITC anti–human IgG Fc secondary antibody. Scale bar = 50 μm. (**C**) Flow cytometry analysis of J87–Dxd internalization at indicated time point. The Jurkat or CCRF–CEM cells only stained with a FITC anti–human IgG Fc secondary antibody served as control. (**D**) Representative scatter plots of 7–ADD (*y*–axis) vs. annexin V (*x*–axis), and the proportion of apoptotic cells are shown in the bar chart (**E**). Data are presented as the means ± SE of triplicate experiments. *** *p* < 0.001.

**Figure 4 biomolecules-14-00106-f004:**
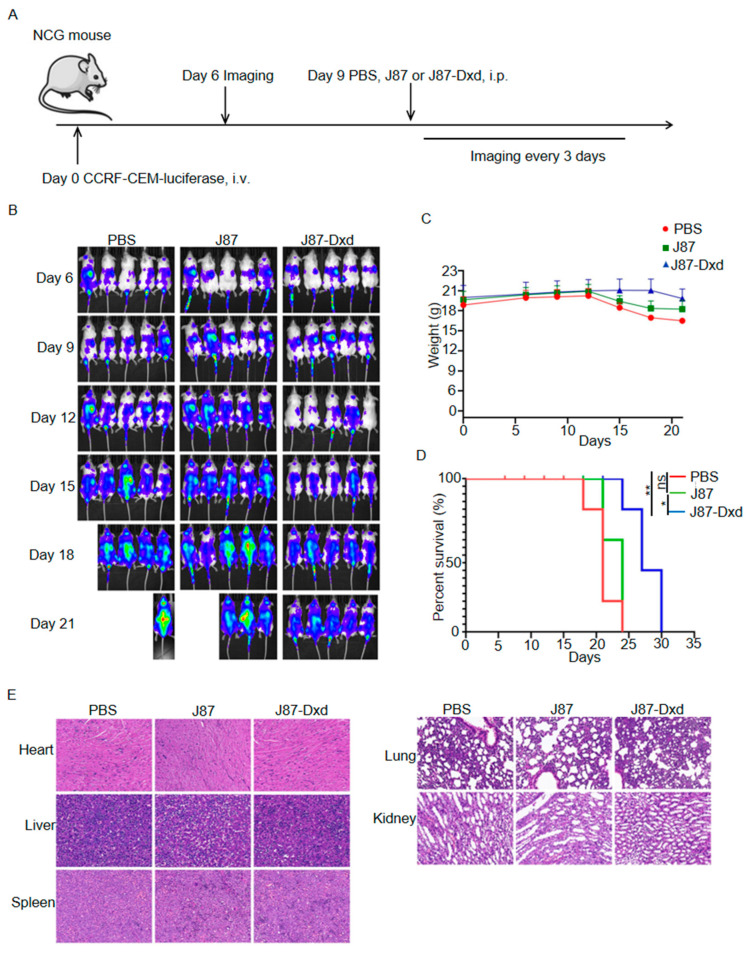
Generation and cytotoxicity assessment of J87-Dxd. (**A**) Treatment scheme used in the CCRF-CEM-luciferase cells. (**B**) After 6 days of tumor injection, an IVIS imaging system was used to monitor tumor growth every 3 days (5 mice per group). (**C**) The weights of mice in different groups. (**D**) Overall survival of mice in each group. * *p* < 0.05; ** *p* < 0.001. ns represents no significance. (**E**) Representative images of H&E staining of the indicated organs of mice in each group.

**Table 1 biomolecules-14-00106-t001:** Biacore kinetic and affinity determinations on mAbs.

mAbs	Ka (1/Ms)	Kd (1/s)	KD (M)
J87	1.21 × 10^6^	1.87 × 10^−4^	1.54 × 10^−10^
G73	1.16 × 10^6^	3.41 × 10^−4^	2.94 × 10^−10^
A15	7.9 × 10^5^	2.5 × 10^−4^	3.17 × 10^−10^

## Data Availability

The original data can be requested to authors.

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
