# Peer review of "An Anti-CD7 Antibody–Drug Conjugate Target Showing Potent Antitumor Activity for T-Lymphoblastic Leukemia (T-ALL)"

_biomolecules, 2024, doi:10.3390/biom14010106_

Round 1

Reviewer 1 Report

Comments and Suggestions for Authors

The works describes the generation of an ADC against the CD7-target expressed in human T and natural killer cells. The work includes the early stage development from immunization of the target in mice, Ab selection, chimerization, functionalization and in vivo and in vitro proof of concept for efficacy.

In order to make the work verifiable and usable for the scientific community, the sequence would have to be published. Furthermore, quality control assays of the antibody and the ADC are missing (identity, integrity, epitope, DAR, disulfide analysis after functionalization). The proof of the selectivity for tumor cells is not convincing.

Comments on the Quality of English Language

Row 15                                 we obtained

Row 26                                inoculationall

Row 31                                expression

Reference required for „Although various therapeutic modalities, including chemotherapy, CAR-T therapy and bone-marrow transplan- 51 tation are applied to the T-ALL, most of them have serious adverse effects or restrictions 52 of use.”

Row 71                                acting as

Sequence CD7

Row 115                              domain not domian

Row 115                              derived

Row133                              stayed

Row 135                              revise: Then centrifuge at…and desalt…

Murine antibody isotype ??

Coupling density?

Open disulfide bridges ? Scrambling ?

SDS-PAGE: adapt MW range to intact Abs

Row 206:            were used as primary antibodies, respectively,

Figure 2A                            legend explaining the color.

Figure 2C                             include x- and y- axis label; 0 and 6 h almost the same color, include cell type in figure text

Row 222                              was instead of is

Figure A                               “imaging every 3 days”

Row 281                              since

Explanation for                 “CD7 is also extensively expressed in normal T cells and natural killer cells” and no side effects ?

Row 287                              anti-HEAR2 ADC

Row 291                              that that

Row 292                              obtained

Row 296                              indicating

Row296                              remained kept

Row 302-306                    simple repetition of results – no discussion ?

Row 307                              our data

References                          inconsistent, incomplete references

Author Response

On behalf of the authors, I would like to express my deep appreciation for your valuable comments on our manuscript. The corresponded responds have been provided in the word document. Thanks for you again.

Reviewer 2 Report

Comments and Suggestions for Authors

This manuscript presents an intriguing study on evaluating the J87-Dxd ADC as a potential new therapy for T-ALL and other CD7-expressing tumors. The theme is certainly compelling; however, I have identified several areas that require further clarification or additional data:

  1. DAR Analysis: The manuscript lacks detailed information on the ADC's synthesis, specifically the DAR. It is critical to include analytical methods and data for DAR, possibly through HIC or RP-HPLC.

  2. Developability of the Antibody: Commentary on the developability aspects of the antibody used in this ADC would be beneficial. This includes its stability, manufacturability, and any potential immunogenicity.

  3. Comparative Analysis in In Vitro and In Vivo Studies:
    For a more comprehensive evaluation, it would be advantageous to compare the J87-Dxd ADC's efficacy with an isotype ADC in both in vitro and in vivo settings. This comparison could provide more insight into the specificity and potency of the J87-Dxd ADC.

  4. Internalization Mechanism:
    The manuscript concludes that antigen-mediated internalization is the primary mechanism. However, have you considered the potential for internalization driven by Fc functions? Including an evaluation of an Fc-silent mAb-derived ADC might offer valuable insights into the internalization mechanisms at play.

  5. Additional Pharmacokinetic, Toxicology, and Plasma Stability Evaluations: It would be beneficial to include data on pharmacokinetics, toxicity, and plasma stability of the ADC. These parameters are essential to understand the therapeutic window, safety profile, and in vivo stability of J87-Dxd.

I believe that addressing these points will significantly strengthen the manuscript and provide a more robust understanding of the J87-Dxd ADC’s potential as a therapeutic agent.

Author Response

(The authors gave the same response as above.)

Reviewer 3 Report

Comments and Suggestions for Authors

Summary:

The manuscript by Shiqi Wang et al. presents a study in which they isolated three monoclonal antibodies (J87, G73, and A15), with J87 exhibiting the highest affinity for CD7. The authors then synthesized an antibody-drug conjugate with deruxtecan (J87-Dxd) and assessed its anti-tumor activity against Acute T-lymphoblastic leukemia (T-ALL).

Review:

This paper demonstrates that the antibody-drug conjugate (J87-Dxd) was found to be more potent for T-ALL. The work signifies an advancement in antibody-drug conjugates and is of interest to researchers in the fields of drug discovery and cancer therapy. However, there are some questions and concerns listed below that should be addressed before publication.

1.     The control of deruxtecan should also be included in the mice studies.

2.     Rephrase this line in Abstarct “Like J87, J87-Dxd remained the ability of recognizing and binding CD7”.

3.     Mice treated with J87 or PBS died. (J87 died or PBS).

4.     Typographical error in abstract other CD7-expression (exprssion)

5.     Typographical error on page 1 line 40 (adults with ALL far (fare) worse…)

6.     Page 2 line 56 says eleven ADCs have been approved by FDA (it should be thirteen https://link.springer.com/article/10.1007/s40290-023-00505-8)

7.     Typographical error on Page 8 line 239 7-weeks-old (7-year-old)

8.     Typographical error on Page 8 line 241 3 groups of five mice each 5 mice.

Also please use high-resolution images 1D, 3A, IC50 curves

Comments on the Quality of English Language

It needs improvement for a better understanding. Some lines are confusing like Like line 20 in the abstract., Line 76 and others.

Author Response

(The authors gave the same response as above.)

Reviewer 4 Report

Comments and Suggestions for Authors

In this manuscript, the researchers identified three anti-CD7 antibodies and subsequently conjugated one of them with drug Dxd, demonstrating its anti-tumor activity against T-ALL. While the study design is straightforward, the efficacy of the antibody-drug anti-tumor function appears to be relatively weak. Several comments are provided below for the improvement of this study.

1. Figure 1A should feature a distinct band representing the CD7 protein to enhance clarity.

2. It is recommended that the authors include the amino acid sequence of the antibody in the supplementary document, particularly focusing on the antibody family and the amino acid sequences of the heavy chain and light chain CDR3. This additional information will contribute to a more thorough comprehension of the study and assist in further analyses.

3. In Figure 4D, the inclusion of a group treated with Dxd alone is essential for a comprehensive comparison and evaluation of the anti-tumor effects. This additional group will provide a baseline reference and enhance the overall robustness of the study findings.

Author Response

On behalf of the authors, I would like to express my deep appreciation for your valuable comments on our manuscript.

  1. Figure 1A should feature a distinct band representing the CD7 protein to enhance clarity.

Resond 1: Thanks. Both our CD7 protein band and the CD7 band shown on the Yeasen official website (https://www.yeasen.com/products/detail/3826) are distinct. The appearance may be attributed to extensive glycosylation of CD7 protein.

  1. It is recommended that the authors include the amino acid sequence of the antibody in the supplementary document, particularly focusing on the antibody family and the amino acid sequences of the heavy chain and light chain CDR3. This additional information will contribute to a more thorough comprehension of the study and assist in further analyses.

Respond 2: Thanks for your valuable suggestions. Thanks for your valuable suggestions. Currently, the writing of the relevant patent has not been completed. In addition, we will publish an article on CD7-targeted CAR-T cell therapy, in which the sequences of anti-CD7 mAbs will also be published.

       3. In Figure 4D, the inclusion of a group treated with Dxd alone is essential for a comprehensive comparison and evaluation of the anti-tumor effects. This additional group will provide a baseline reference and enhance the overall robustness of the study findings.

Respond 3: Thanks for your valuable suggestions. In fact, in our previous experiments, we treated mice only with Dxd (10 μg/kg), but the survival time of Dxd-treated mice was not prolonged compared to PBS treatment.

Round 2

Reviewer 2 Report

Comments and Suggestions for Authors

No further comments are needed

Author Response

Thank you for taking your precious time to review our manuscription. Maybe, the options located at Bottom of the page need your confirmation (Are the methods adequately described?). Thanks again for your valuable suggestions. 

Reviewer 4 Report

Comments and Suggestions for Authors

In Figure 4D, the inclusion of a group treated with Dxd alone is essential for a comprehensive comparison and evaluation of the anti-tumor effects. This additional group will provide a baseline reference and enhance the overall robustness of the study findings.

The author should include this control group in the figure if they have the data already.

Author Response

In Figure 4D, the inclusion of a group treated with Dxd alone is essential for a comprehensive comparison and evaluation of the anti-tumor effects. This additional group will provide a baseline reference and enhance the overall robustness of the study findings.

The author should include this control group in the figure if they have the data already.

Respond: Thanks for your valuable suggestions. To exclude the possibility that this antitumor effects were generated by free Dxd, we also performed an experiment to compare the survival time of CCRF-CEM mice treated with Dxd alone with PBS vehicle control. As a result, No significant difference were observed between the Dxd- and PBS-treated mice. We have added this result in Figure S3 according to your suggestion. The relevant descriptions were highlighted in red.